# Psychosocial correlates of safe sex self-efficacy among in-school adolescent girls in Lagos, Nigeria

Ucheoma Nwaozuru[1]*, Sarah Blackstone[2], Chisom Obiezu-Umeh[1], Donaldson F. Conserve[3], Stacey Mason[1], Florida Uzoaru[1], Titi Gbajabiamila[4], Oliver Ezechi[4], Patricia Iwelunmor[5], John E. Ehiri[6], Juliet Iwelunmor[1]

1 Department of Behavioral Science and Health Education, College for Public Health and Social Justice, Saint Louis University, Saint Louis, Missouri, United States of America, 2 Departments of Health Professions and Health Sciences, James Madison University, Harrisonburg, Virginia, United States of America, 3 Department of Health Promotion, Education, and Behavior, University of South Carolina, Columbia, South Carolina, United States of America, 4 Nigerian Institute of Medical Research, Yaba, Lagos State, Nigeria, 5 Morning Star Health and Human Development Foundation, Festac Town, Lagos State, Nigeria, 6 Department of Health Promotion Sciences, Mel and Enid Zuckerman College of Public Health, University of Arizona, Tucson, Arizona, United States of America

* nwaozur2@gmail.com

**Data Availability Statement:** All data files are available from the OPENICPSR databases (accession number: openicpsr-117403)

# Abstract

## Background

Adolescent girls in Nigeria are at heightened risk for HIV and other sexually transmitted infections. However, there are limited studies on psychosocial factors that are associated with safe sex intentions among this population. Self-efficacy has been established as an important correlate of behavioral intentions and the actual behavior. The objective of this research was to examine how key psychosocial factors such as social support, parental monitoring, and future orientation influence perceived safe sex self-efficacy among in-school adolescent girls in Nigeria. Furthermore, we assessed the associations between these psychosocial factors and HIV-related knowledge and safe sex self-efficacy.

## Methods

A self-administered questionnaire was distributed to 426 adolescent girls attending public and private school systems in Lagos, Nigeria. Multiple linear regression was used to evaluate the influence of psychosocial and demographic factors on safe sex self-efficacy. Further, stratified analysis was conducted to compare the estimates between participants attending public schools (n = 272) and those attending private schools (n = 154).

## Findings

Results from the study show that future orientation (β = 0.17; p < 0.05), participants age (β = 0.14; p < 0.05), and HIV knowledge accuracy (β = 0.17; p < 0.05) were associated with safe sex self-efficacy. Future orientation remained statistically significant in the sub-group analysis among participants attending public (β = 0.13; p < 0.05) and private schools

**Funding:** The authors received no specific funding for this work.

**Competing interests:** The authors have declared that no competing interests exist.

**Abbreviations:** CO, Chisom Obiezu-Umeh; DC, Donaldson Conserve; FU, Florida Uzoaru; HIV, Human Immunodeficiency Virus; JEE, John E. Ehiri; JI, Juliet Iwelunmor; LGA, Local Government Area; OE, Oliver Ezechi; PI, Patricia Iwelunmor; RAA, Reasoned Action Approach; SB, Sarah Blackstone; SD, Standard Deviation; SM, Stacey Mason; STIs, Sexually Transmitted Infections; TG, Titi Gbajabiamila; UN, Ucheoma Nwaozuru; VIF, Variance Inflation Factor.

(β = 0.24; p < 0.05). Among participants attending public schools, HIV accuracy (β = 0.2; p < 0.05) remained a significant correlate of safe sex self-efficacy while this association dissipated among private school attendees.

## Conclusions

These findings point to the importance of including future orientation strategies in interventions developed for in-school adolescent girls in Nigeria. School-based interventions that increase positive future orientation outcomes may be beneficial to improve safe sex intentions among adolescent girls in Nigeria.

## Introduction

Adolescent girls and young women (AGYW) aged 14–24 are a priority population for Human Immunodeficiency Virus (HIV) prevention in sub-Saharan Africa (SSA), having HIV infection rates twice as high as their male peers [1–3], and accounting for 31% of new HIV infections in the region [3, 4]. In Nigeria, an estimated 12,000 adolescent girls aged 15–19 were newly infected with HIV in 2013 [5]. Recent policy documents in the country (Revised National HIV and AIDS Strategic Framework 2019–2021) also emphasize the growing burden of HIV among this population [6, 7]. The majority of sexually transmitted infections (STIs), including HIV, among AGYW in SSA is attributed to having older sexual partners, unprotected sexual encounter, transactional sex, early sexual debut, and multiple sex partners [8]. In Nigeria, 62.8% of sexually active adolescent girls between 15–19 years report no condom use during their last intercourse, and 39% had sexual intercourse with male partners who were 10 or more years older in exchange for items such as food, money, school fees, and gifts [9]. Although adolescents in Nigeria possess considerable knowledge of the causes, transmission, or prevention of HIV infection, their adherence to HIV-related preventive measures is uncorrelated. A survey-based study of 865 sexually active adolescents in Nigeria suggested that while 75% had good knowledge of HIV, this knowledge did not necessarily translate into HIV protective attitudes [10]. Instead, the cultural construction and social organization of gender in Nigeria that disempowers girls, play a role in increasing their vulnerability to risky sexual behaviors [11]. These factors operate through early marriage, sexual violence, and economic inequities, such as lower wages, limited employment opportunities [12], influencing their sexual behavioral intentions and behaviors. To design interventions tailored to address the unique needs of AGYW, there is need to gain further insight on sexual behavioral intentions among adolescent girls in Nigeria given that behavioral decisions are made based on rational considerations derived from available information and skills.

Establishing a foundation that promotes safe sex self-efficacy, particularly among women, is a fundamental stepping stone to promote safe sexual behavioral intentions. Self-efficacy, defined as having confidence in one's ability to perform a particular behavior, is an important correlate of behavioral intentions and the actual behavior [13]. It reflects an individual's control of, or the skills to execute a specific health behavior; It is an integral predictor of health behavioral intentions and actual execution of the behavior [14]. Self-efficacy is shaped by social norms, knowledge, outcomes expectations, and communication with family, peers, and community members [13]. Evidence from sub-Saharan Africa suggests that self-efficacy skills have the potential to influence adolescent's HIV-prevention practices (i.e. condom use) [15, 16].

However, how self-efficacy skills may increase confidence to engage in risk-reduction behaviors in Nigeria remains unclear.

Other key psychosocial factors that may influence safe sex intentions and reduce risky sexual behaviors among adolescents include social support, parental monitoring, and future orientation [13, 17–20]. A growing number of studies underscore the importance of social support for reducing sexual risk intentions and behaviors among adolescents and young people [16, 21–23]. Studies from South Africa and Uganda suggest that social support is instrumental for reductions in sexual debut, unintended pregnancy, age-disparate sex, and transactional sex [21, 22]. Previous research also indicates that familial factors, particularly parental monitoring, are critical in keeping adolescents safe [19, 24, 25]. Adolescent girls who perceive that their parents or guardians are concerned about their locations and social interactions are substantially less likely to engage in sexual risk behaviors [24, 25]. Finally, future orientation, or the degree to which adolescents possess positive attitudes towards their future, has also been shown to influence sexual behavior and sexual behavioral intentions [26, 27]. Higher future orientation among adolescents has been found to yield higher confidence in engaging in positive health behaviors [28, 29]. Studies suggest that individuals who are pessimistic about their future are more likely to take risks with their health and safety, whereas those with a positive future orientation are less likely to engage in unsafe sexual practices [26, 27]. Despite the influence of these psychosocial factors on sexual behavioral patterns and the high risk of acquiring HIV among adolescents, there are only a few studies in Nigeria that examine how these psychosocial factors influence safe sex self-efficacy among adolescent girls [15, 16].

In light of the aforementioned gaps, we grounded our study design and measures in the Reasoned Action Approach (RAA) which provides a theoretical underpinning to better capture the dynamic between sexual self-efficacy and other psychosocial factors [30, 31]. RAA posits that people's behavioral intentions are formed based on three cognitive constructs: attitudes, perceived norms and behavioral control [30]. The theory conceptualizes why individuals decide to engage or not engage in a particular behavior [30]. RAA has been largely employed to access predictors of behavioral intentions such as sexual self-efficacy. The theory recognizes the relevance of interpersonal processes such as social support, parental monitoring and personal traits such as future orientation and HIV knowledge skills on the development of safe sex self-efficacy [30, 31].

To this end, this study examined the role of social support, future orientation, and parental monitoring may play with influencing safe sex self-efficacy of in-school adolescent girls in Nigeria. It explored the correlates of participants' confidence in their intentions to engage in safe sexual behaviors. Schools were selected because they provide an optimal opportunity to reach many adolescents and a large number of adolescents attend schools before they are predisposed to sexual behaviors that may put them at risk of HIV infection [32]. It is also an ideal platform to implement interventions that build confidence in health-seeking behaviors, thereby enhancing overall self-efficacy [33]. In Nigeria, variations exist between the two main school systems (private and public schools). These schools differ with the availability of financial and human resources [34]. For instance, private schools are more likely to have access to funding because they are mostly for-profit. The present study assessed the psychosocial correlates of safe sex practices among in-school adolescent girls in Nigeria [35].

## Methods

### Study setting and participants

This study utilized a cross-sectional design conducted among female students attending secondary schools in Lagos, Nigeria. Lagos is situated in the Southwest region of Nigeria,

bordered by the Atlantic Ocean. It is the most populous city in Africa with an estimated population of 17 million people out of a national estimate of 150 million in Nigeria [36]. The participants for this study consisted of adolescents in secondary schools from two educational districts in Lagos state (District V and VI). The schools were selected by a simple random method. A list of schools in the two districts were obtained and stratified by type of school (i.e. public-owned and private-owned). We randomly selected 10 schools from each district using random numbers generated with MS Excel. In total, we contacted 20 schools, and participation in the study was based on approval by the school principals. Principals at four schools (3 public and 1 private secondary schools) in District V and eight schools (3 public and 5 private secondary schools) in District VI approved for the study to be conducted at their schools. Altogether, 426 adolescent girls from six public schools (n = 272) and six private schools (n = 154) were recruited for this study. Study participants were eligible to participate in the study if they were 13 years and above, attending secondary schools enrolled in JSS1, JSS2, JSS3, SS1, SS2, and SS3 (equivalent to Grades 6 through 12 in the United States), and able to provide informed assent and parental informed consent.

## Procedures

Ethical approval was obtained from the Institutional Review Board at Saint Louis University and the Nigerian Institute of Medical Research. Before study recruitment, we obtained clearance from Education district VI and V- the school districts for the participating schools. Also, the school principals were contacted individually to further explain the details of the study (ethical approvals and request letters were submitted in each school) and following these discussions approval for the study was given. After the school principals provided written approvals and reviewed the study questionnaire, we distributed letters of intent, written in English, to adolescent girls in the twelve schools to notify their parents of the study. Parents and adolescents at the schools were informed that participation was voluntary and that their responses to the study survey were confidential and anonymous. Students who were under 17 years and interested in the study were provided with a consent form to take home to their parents before the study date to obtain their parents' consent. All participants who met the study criteria and were interested in the study were invited to converge in a classroom at each school for the study. At the beginning of each research meeting, written consents were obtained from all study participants before participation in the study. For participants aged 17 years and under, we obtained written youth assent and parent/guardian consent before participation in the study. Then, an anonymous paper survey questionnaire designed by the authors was administered to adolescent girls. The questionnaires were self-administered and completed using a pen or pencil. The survey instruments were administered in English, being that English is the national language in Nigeria. The questionnaire took approximately forty-five minutes to one hour to complete. The research protocol was approved by the Institutional Review Board at Saint Louis University and the Nigerian Institute of Medical Research.

**Dependent variable.**  *Safe sex self-efficacy*. A 15-item self-efficacy scale adapted from self-efficacy scales by Grimley et al. [37] and Parsons et al. [38] were used to assess participants' safe sex self-efficacy. The scale measured adolescent girls' confidence in their intentions to say "no" to sexual intercourse with their partners in a variety of situations such as partners whose sexual relationship history is not known to them or partners whom they have known for a few days or less. The scale also measured adolescent girls' confidence in their intentions to use condoms consistently with their partners. An example of the scale item included, "I feel confident that I can say no to sex if my partner refused to use a condom." Items were rated on a 4-point

Likert Scale (1 = Not sure at all; 2 = A little sure; 3 = Sure; 4 = Very sure). Higher scores indicated greater self-efficacy. Possible scores ranged from 4 to 60, (Cronbach's α = .85).

**Independent variables.** Independent variables were considered based on the guiding theory (RAA) and prior empirical evidence on psychosocial factors that influence safe sex self-efficacy [13, 17–20]. Responses for the measures were reverse coded where necessary to ensure that higher values reflected greater levels of a given scale.

*Future orientation*. Future orientation was assessed with 10 questions adapted from Whitaker et al. [39] and Cabrera et al. [26] which addressed attitudes about the future. Participants were asked to respond to a 4-point Likert-type scale ranging from 1 = never true to 4 = always true to statements such as "what happens to me in the future mostly depends on me" and "It's really no use worrying about the future because what will be will be." Future orientation scores were computed such that higher scores indicated greater levels of certainty regarding future plans. Possible scores ranged from 4 to 40, (Cronbach's α = .52). In no case did the deletion of an item increase the α by 0.01.

*Perceived social support*. We assessed perceived social support using 8-items adapted from Stewart et al. [40] 12-item Multidimensional Scale of Perceived Social Support (MSPSS). This scale provides a brief measure of social support designed to measure respondents' perception of the adequacy of the support she receives. The scale included items such as "I feel like I can talk to my family about my problems," with responses rated on a 4-point Likert-type scale ranging from 1 = strongly disagree to 4 = strongly agree, with higher scores indicating a greater sense of community and social support. Possible scores ranged from 4 to 32, (Cronbach's α = .66). In no case did the deletion of an item increase the α by 0.01.

*Parental monitoring*. Five items adapted from Vieno et al. [41] was used to assess parental monitoring of adolescent girls' leisure time after school. The scale items included questions such as "How much does your mother/father/guardian really know about who your friends are?" Responses were rated on a 3-point scale (1 = knows a lot; 2 = knows a little; 3 = doesn't know anything). All questions were recoded so that higher scores on this sub-scale indicated greater parental monitoring of activities. Scores ranged from 5 to 15 (Cronbach's α = .68). In no case did the deletion of an item increase the α by 0.01.

*HIV knowledge*. We used the 23-item AIDS Risk Knowledge Test survey by Kelly et al.[42] which contains true/false/don't know statements assessing practical knowledge of AIDS risk behavior. "Do not know responses" were treated as incorrect responses in computing the scale scores. Higher scores on this subscale indicated greater accuracy in HIV/AIDS knowledge. Possible scores ranged from 0 to 23. The Cronbach's alpha estimate of internal consistency was 0.75 [43].

**Covariates.** Demographic Factors: We assessed participants' age (continuous variable in the analyses), ethnicity (categorized as Igbo, Yoruba, Hausa, and Others), educational level (categorized as JSS1, JSS2, JSS3, SS1, SS2, and SS3), religion (Categorized as Christianity or Islam), and whether they had a mentor (Yes or No response). This was defined as someone over 25 years that they could go to for support and guidance if they needed to make an important decision, or who inspired them to do their best. We also assessed school type (public or private schools).

## Data analysis

All statistical analyses were conducted using the Statistical Package for the Social Science (SPSS) version 25. We conducted descriptive analyses of participants' demographic characteristics, safe sex self-efficacy, future orientation, perceived social support, parental monitoring, and HIV knowledge accuracy. Bivariate analyses were conducted to examine whether there

were differences in safe sex self-efficacy across participants' demographic factors. This was accomplished through using a) Pearson product-moment correlation to establish whether age and safe sex self-efficacy were associated, b) Spearman's rank-order correlation to establish whether participants' education level and safe sex self-efficacy were associated, and c) bivariate linear regression was used to examine associations between participants' ethnicity, religion, having a mentor, type of school, and safe sex self-efficacy, respectively.

We further compared mean scores for safe sex self-efficacy, future orientation, perceived social support, parental monitoring, and HIV knowledge accuracy. T-tests were used to confirm if these factors differed significantly between attendees of public versus private schools.

Finally, we examined whether demographic factors (age, education, ethnicity, religion, mentorship), future orientation, perceived social support, parental monitoring, and HIV knowledge accuracy influenced safe sex self-efficacy in multiple linear regression. Three separate analyses were conducted. First, the influence of demographic and psychosocial factors on safe sex self-efficacy were examined in the entire sample (n = 426). Subsequently, we ran two additional models, one with public-school attendees (n = 272) and one with private school attendees (n = 154). Multicollinearity among the main effects of the independent variables and covariates were assessed in the regression model using the variance inflation factor (VIF) value of multicollinearity tests. Statistical significance was defined at the level of a p-value equal to or less than 0.05.

## Results

### Participants characteristics

The study sample consisted of 426 adolescent girls and young women between 13 and 19 years. The average age of participants was 15.02 years (*SD = 1.21)*. The majority of the participants (32.6%) were 15 years, and only one participant (0.2%) was 19 years. Nearly 43% (*n* = 184) of the participants were in Senior Secondary class 2 (SS2). In terms of ethnicity, 169 participants (39.2%) were Igbo, 152(35.7%) were Yoruba, and 16 (3.8%) were Hausa. The remaining 15% (*n* = 64) were from other ethnic groups. Over three-fourth (*n* = 333) of participants were Christians and the remaining one-fourth (*n* = 86) were Muslims. Three hundred and thirty-seven participants (79.1%) reported that they believed they had a mentor. Mentors included parents (46.59%, *n* = 157), siblings (19.29%, *n* = 65), extended family including uncles, aunts, and cousins (17.51%, *n* = 59) and non-family relations including guardians, religious leaders, school teachers, neighbors and older friends (16.62%, *n* = 56). Two hundred and seventy-two participants (63.8%) attended public schools. Table 1 presents further descriptive demographic characteristics of study participants and their association with safe sex self-efficacy from bivariate analyses.

The bivariate relationship between participants' sociodemographic characteristics and safe sex self-efficacy are also presented in the column labeled *"Test of Association"* in Table 1. The results of the bivariate analysis show a relatively weak, positive association between participants' age and safe sex self-efficacy scores, *r* (426) = 0.115, *p*<0.05. However, there were no statistically significant associations between any of the other sociodemographic characteristics (participants' education level, ethnicity, religion, type of school, and having a mentor) and safe sex self-efficacy (*p*>0.05).

### Descriptive statistics of independent variables

Overall, participants demonstrated relatively high levels of safe sex self-efficacy, future orientation, social support, parental monitoring, and HIV knowledge accuracy. For most of the scales, the mean score is above the midpoint of the response scale. However, for HIV knowledge

**Table 1. Selected characteristics of in-school adolescent girls in Lagos, Nigeria, and bivariate relationship with safe sex self-efficacy (N = 426).**

| Participants Characteristics | n (%) | Test of Association |
|---|---:|---:|
| Age, mean (SD) | 15.02 (1.21) | r = 0.118* |
| 13 years | 30 (7.0) | |
| 14 years | 128 (30.0) | |
| 15 years | 139 (32.6) | |
| 16 years | 78 (18.3) | |
| 17 years | 35 (8.2) | |
| 18 years | 15 (3.5) | |
| 19 years | 1 (0.2) | |
| Education | | r = 0.061 |
| JSS1 | 1 (0.2) | |
| JSS2 | 2 (0.5) | |
| JSS3 | 10 (2.3) | |
| SS1 | 159 (37.3) | |
| SS2 | 184 (43.2) | |
| SS3 | 62 (14.6) | |
| Ethnicity | | F = 0.746 |
| Igbo | 167 (39.2) | |
| Yoruba | 152 (35.7) | |
| Hausa | 16 (3.8) | |
| Other | 64 (15.0) | |
| Religion | | F = 0.389 |
| Christianity | 333 (78.2) | |
| Islam | 86 (20.2) | |
| Mentor | | F = 0.286 |
| Yes | 337 (79.1) | |
| No | 85 (20.0) | |
| Type of School | | F = 0.365 |
| Private | 154 (36.2) | |
| Public | 272 (63.8) | |

*p < .05,

** p < .01,

*** p < .001.

JSS = Junior Secondary School; SS = Senior Secondary; SD = Standard deviation

Data are number (percent) of participants unless otherwise indicated. For some of the characteristics, the totals may differ from the participants' totals owing to missing data.

accuracy, the mean score is below the response scale. The scores on safe sex self-efficacy ranged from 4 to 60, with a mean of 42.14 (SD = 11.38) (See Table 2). Comparisons for independent variable scales between attendees of public and private schools are also presented in Table 2. There were no statistically significant differences in safe sex self-efficacy, future orientation, social support, parental monitoring, and HIV knowledge accuracy scores among individuals in public versus those in private schools (p>0.05). Although not statistically significant, participants in public schools had higher mean scores on safe sex self-efficacy and social support. Participants in private schools had greater mean scores on future orientation, parental

**Table 2. Descriptive statistics of psychosocial scales and HIV knowledge accuracy among in-school adolescent girls in Lagos, Nigeria, (N = 426).**

| | Mean Total (SD) | Public School Attendees | | | | Private School Attendees | | | | F | P-Value |
|---|---|---|---|---|---|---|---|---|---|---|---|
| | | Mean | SD | Minimum | Maximum | Mean | SD | Minimum | Maximum | | |
| Safe sex self-efficacy | 42.14 (11.38) | 42.4 | 11.7 | 4 | 60 | 41.7 | 10.82 | 4 | 60 | 0.37 | 0.55 |
| Future orientation | 31.96 (5.19) | 31.76 | 5.53 | 4 | 40 | 32.32 | 4.55 | 16 | 40 | 1.12 | 0.29 |
| Social Support | 25.57 (4.51) | 25.75 | 4.53 | 5 | 32 | 25.28 | 4.48 | 4 | 32 | 1.02 | 0.31 |
| Parental Monitoring | 12.29 (2.23) | 12.28 | 2.24 | 5 | 15 | 12.29 | 2.22 | 5 | 15 | 0.003 | 0.96 |
| HIV Knowledge accuracy | 10.1 (3.87) | 10.01 | 3.92 | 0 | 20 | 10.23 | 3.79 | 0 | 21 | 0.3 | 0.58 |

monitoring and HIV knowledge accuracy than those in public schools; however, as stated earlier. these differences were not statistically significant.

## Intercorrelations of safe sex self-efficacy and independent variables among in-school adolescent girls in Lagos, Nigeria (N = 426)

Table 3 presents the Pearson product-moment correlation between safe sex self-efficacy and independent variables (future orientation, social support, parental monitoring, and HIV knowledge accuracy). Safe sex self-efficacy was related significantly with HIV knowledge accuracy scores ($p<0.001$) and future orientation scores ($p<0.01$). No statistically significant relationships were found between safe sex self-efficacy, social support, and parental monitoring ($p<0.05$). Also, social support was significantly related to parental monitoring and future orientation scores.

## Multiple linear regression analysis

Table 4 presents the results of the multiple linear regression, examining the influence of psychosocial factors (parental monitoring, social support, and future orientation) and HIV knowledge accuracy on safe sex self-efficacy for all participants, and separated by public and private schools. The regression models controlled for participants age, education level, religion, ethnicity, and having a mentor. Multicollinearity among the main effects of the independent variables and covariates was assessed in the regression model using the VIF value of multicollinearity tests. Based on the VIF values (range: 1.06–1.70 for the three models) there were no multicollinearity concerns identified.

Overall, future orientation ($\beta = 0.17$; $p < .05$), HIV accuracy ($\beta = 0.17$; $p < .05$) and participants age ($\beta = 0.14$; $p < .05$) were positively associated with safe sex self-efficacy. Higher scores on future orientation and HIV accuracy were associated with increased safe sex self-efficacy

**Table 3. Correlations of independent variables and safe sex self-efficacy.**

| | | 1 | 2 | 3 | 4 | 5 |
|---|---|---|---|---|---|---|
| 1 | Parental monitoring | 1 | 0.076 | 0.259*** | 0.088 | 0.074 |
| 2 | HIV Accuracy | 0.076 | 1 | -0.021 | 0.188*** | 0.192*** |
| 3 | Social Support | 0.259*** | -0.021 | 1 | 0.164** | 0.029 |
| 4 | Future orientation | 0.088 | 0.188*** | 0.164** | 1 | 0.193** |
| 5 | Safe sex self-efficacy | 0.074 | 0.192*** | 0.029 | 0.193** | 1 |

*$p < .05$,

** $p < .01$,

*** $p < .001$

**Table 4. Association between the psychosocial factors, sociodemographic factors and safe sex self-efficacy among in-school adolescent girls in Lagos, Nigeria, (N = 426).**

| Predictors | Overall | | | Public school | | | Private school | | |
|---|---|---|---|---|---|---|---|---|---|
| | β | SE | t | β | SE | t | β | SE | t |
| Future orientation | 0.17** | 0.2 | 3.18 | 0.13* | 0.14 | 2.00 | 0.24** | 0.19 | 2.82 |
| HIV Accuracy | 0.17*** | 0.15 | 3.31 | 0.2** | 0.2 | 2.94 | 0.14 | 0.23 | 1.63 |
| Social Support | 0.04 | 0.15 | 0.8 | -0.009 | 0.19 | -0.13 | 0.13 | 0.23 | 1.53 |
| Parental Monitoring | 0.05 | 0.27 | 1.02 | 0.08 | 0.36 | 1.16 | -0.001 | 0.41 | -0.016 |
| Age | 0.14* | 0.53 | 2.33 | 0.14 | 0.71 | 1.85 | 0.11 | 0.9 | 1.04 |
| Mentor | 0.07 | 1.47 | 1.27 | 0.08 | 1.99 | 1.16 | 0.06 | 2.18 | 0.71 |
| Education | -0.02 | 0.82 | -0.32 | -0.05 | 1.16 | -0.65 | 0.03 | 1.26 | 0.28 |
| Religion | 0.001 | 1.49 | 0.03 | 0.07 | 1.81 | 1.08 | -0.16 | 2.8 | -1.97 |
| Ethnicity | 0.03 | 0.54 | 0.58 | -0.03 | 0.8 | -0.41 | 0.11 | 0.72 | 1.37 |

*$p < .05$,

** $p < .01$,

*** $p < .001$.

*Note*: β = standardized regression coefficients; *SE* = Standard error

scores. Similarly, Higher age groups were associated with increased safe-sex self-efficacy scores. In public schools, only future orientation ($β = 0.13$; $p < .05$) and HIV accuracy ($β = 0.2$; $p < .05$) remained significantly associated with safe sex self-efficacy. Higher levels of future orientation and HIV accuracy were positively associated with safe sex self-efficacy. In private schools, future orientation remained a significant correlate of safe sex self-efficacy ($β = 0.24$; $p < .05$); higher levels of future orientation were associated with greater safe sex self-efficacy. Also, religion was significantly associated with safe self-efficacy among participants in private schools ($β = 0.11$; $p < .05$). Participants identifying as Muslims were more likely to have higher levels of safe sex self-efficacy in private schools. To further explore this relationship, we conducted a bivariate analysis between religion and safe sex self-efficacy, which was not statistically significant. However, when the bivariate analysis between religion and safe sex self-efficacy was stratified by school type. The association between religion and safe sex self-efficacy was statistically significant for participants in the public school (t = 20.70 (df = 60), p = 0.03) but not statistically significant for participants in private schools (t = 38.40 (df = 141), p = 0.10).

## Discussion

This study extends adolescent research by examining the extent to which psychosocial factors-social support, future orientation, and parental monitoring- influence safe sex self-efficacy among a sample of in-school adolescent girls in Nigeria. As previous studies have shown that HIV knowledge does not necessarily translate to safe sexual behavior [10, 32], guided by the reasoned action approach this study sought to examine how key psychosocial factors beyond HIV knowledge may influence safe sex self-efficacy. Self-efficacy to perform safe sexual behavior is an important predictor of behavioral intentions and actual safe sexual behaviors among adolescents [44, 45]. This is one of the first studies to our knowledge to examine the complex interplay between psychosocial factors and their possible influence on confidence in intentions to engage in safe sexual behaviors among adolescent girls in Nigeria.

Prior studies have reported that adolescent girls reporting more social support, parental monitoring, and positive attitudes about their future are less likely to engage in unsafe sexual

practices [13, 17–19]. However, these studies have not been conducted in Nigeria where 15% of girls are sexually active before the age of 15 [9]. In the present study, we found that participants in private schools did not differ from those in public schools on their belief in their ability to act successfully to prevent HIV. This is represented by their scores on the measure of safe sex self-efficacy. Overall, we found future orientation and HIV knowledge accuracy to be positively associated with safe sex self-efficacy.

Concerning the link between safe sex self-efficacy and psychosocial factors, there was a strong correlation between safe sex self-efficacy and future orientation. The multivariate regression of the independent variables and co-variates also showed that future orientation is an important correlate of safe sex self-efficacy. This is consistent with previous studies that indicate a positive relationship between optimistic future orientation and safe sex self-efficacy among different adolescent populations [28, 46–48]. Individuals, especially vulnerable adolescents, with positive future orientation are less likely to engage in risky health behaviors such as marijuana use, alcohol during sex, and unprotected sex [28]. The potential reason as explained by previous literature is that individuals who have positive outlooks for the future are more likely to take deliberate actions to protect themselves from actions that can jeopardize their future [49–51]. This includes refraining from risky sexual behaviors and self-efficacy for safe sexual practices [49, 50].

Results from the study also suggest older adolescents exhibited higher safe sex self-efficacy compared to younger adolescents. This finding is congruent with other research studies [52–54] that found an association between age and safe sex self-efficacy. In these studies, safe sex self-efficacy increased with age. A possible explanation for this is that older adolescents may have more exposure to HIV knowledge due to more years of schooling and social interaction compared to younger adolescents. This may have increased their confidence in practicing safe sex, as increased HIV knowledge in this study was associated with increases in safe sex self-efficacy. However, age diminishes as an important correlate of safe sex self-efficacy in the subgroup analysis by type of school. This suggests that psychosocial factors such as future orientation may be a more salient correlate of safe sex self-efficacy than age.

We also conducted a subgroup analysis examining the association of psychosocial factors on safe sex self-efficacy in public schools versus private schools. There were no differences in the psychosocial correlates of safe sex self-efficacy among participants attending either public or private schools. Future orientation remained an important correlate of safe sex self-efficacy among adolescent girls in both the public and private secondary schools. This further strengthens the importance of fostering positive future orientation to promote safe sex self-efficacy attitudes as well as safe sexual behaviors among adolescent girls. Interestingly, among participants in private schools, religion was a significant correlate of safe sex self-efficacy, with young women identifying as Muslims showing higher safe sex self-efficacy scores. It is unclear whether girls in private schools reporting to be Muslims have ideas or exposures that increase safe sex self-efficacy compared to non-Muslim public-school girls. Although not measured in this study, some researchers have found an association between strong religious beliefs, or religiosity, among adolescents and safe sex self-efficacy [55] and condom use [56].

It is also noteworthy that HIV knowledge accuracy was associated with greater safe sex self-efficacy. In the subgroup analysis, HIV knowledge was not a significant correlate of safe sex self-efficacy among participants in private schools, but it was a significant correlate among participants in public schools. Thus, while HIV knowledge may not be associated with sexual behaviors, it could potentially be indirectly associated with sexual behaviors and mediated by safe sex self-efficacy. While this study did not examine actual sexual behavior, this potential mediating relationship should be considered in future studies. This relationship may be

different for adolescent girls and young women in Nigeria given the complex interplay of cultural and social factors that increase their vulnerabilities in engaging in risky sexual behaviors [12].

## Implications for interventions and future research

These findings are of practical importance as it provides insights into the role future orientation may play with enhancing safe sex self-efficacy for adolescent girls in school settings. Since orientation toward the future provides the motivation that guides the attainment of goals, appropriate interventions targeted to adolescent girls should include strategies that foster thinking about the future in positive ways [57]. These strategies could enhance their expectations or impressions of what the future holds, aspirations or intentions for the future, and planning or creating awareness on their ability to create a program of action to achieve their aspirations [27]. Future research needs to also explore the contexts that influence future orientation and other psychosocial strategies and not simply assume a common context for all adolescent girls [27]. By taking contextual factors into account, studies may provide insight into how opportunities and constraints present themselves in an environmental context and influence adolescent girls' future thoughts and safe sex self-efficacy. Finally, given that we found a strong association between safe sex self-efficacy and future orientation, interventions that increase adolescent girls' future orientation may be a valuable means for not only reducing risky sexual behaviors but also for promoting a successful transition to adulthood. Altogether, findings from this study may inform the design of interventions that effectively consider the role psychosocial factors play in sexual decision-making,intentions and behaviors among adolescent girls in Nigeria.

## Limitations

There are some limitations to this study worth addressing. First, as with any type of survey research, there is the risk of social desirability and recall bias. Though the participants completed the surveys individually and anonymously, there is still a risk that they altered their responses due to the nature of the study. There is also a risk that participants were not able to recall accurate information for some of the questions, for instance, those regarding parental monitoring. Second, while we used a standardized instrument to measure safe sex self-efficacy, perceived social support and future orientation, the Cronbach alpha on these items were low. Users of these instruments should be aware that the low internal validity of these instruments could affect the reliability of the results. Third, we did not ascertain which of the participants were sexually active. Future studies among adolescent girls can further explore if there are differences in safe sex efficacy among adolescent girls based on their sexual activity status. Fourth, the cross-sectional nature of this study precludes assumptions about causal relationships between the independent variables and the primary outcome, safe sex self-efficacy. Further research is needed to clarify the association between future orientation, social support, parental monitoring, and safe sex self-efficacy among adolescent girls in Nigeria using a longitudinal study design. Lastly, we utilized a convenience sample of adolescent girls attending secondary schools. This may limit our ability to generalize these findings to other adolescent girls, such as out-of-school adolescent girls in Nigeria. However, the demographics of the students who participated in this study are similar to other studies among adolescent girls in Nigeria [58, 59].

Despite these limitations, the strength of this study lies in the fact that it is one of the first to examine the relationship between psychosocial factors and self-efficacy to engage in safe sexual behaviors among adolescent girls in Nigeria, a group that is currently at high risk for HIV

transmission. Most of the questions were regarding participants' current emotional and psychological state, thus limiting the impact of recall bias for these items. The findings are useful for informing interventions that aim to increase safe sex self-efficacy among young women in Nigeria.

## Conclusion

This study highlights the importance of looking beyond HIV knowledge in promoting intentions and actual safe sexual behaviors among adolescent girls in Nigeria. Prior research has shown that knowledge alone does not translate to reduced risky sexual behaviors [10], thus indicating a need to focus on psychosocial factors beyond individual knowledge. The current study enhances knowledge of correlates of safe sex self-efficacy by showing that an individual's positive outlook about the future (i.e. future orientation) is associated with heightened confidence in safe sex intentions. Future interventions can potentially target these factors to improve safe sex self-efficacy to ultimately increase safe sex practices among adolescent girls. Additionally, more studies are needed to examine how the school environment might be best shaped to provide resources and supportive interventions that promote future orientation among adolescent girls in Nigeria.

## Supporting information

**S1 File. Questionnaire used for study data collection.**
(PDF)

## Acknowledgments

The authors wish to thank Morning Star Health and Human Development Foundation and Lagos State School District V and VI for their support in ensuring the success of this study. We also wish to thank all participants in this study for their interest and for their time spent in completing the study questionnaire.

## Author Contributions

**Conceptualization:** Juliet Iwelunmor.

**Data curation:** Ucheoma Nwaozuru, Sarah Blackstone, Juliet Iwelunmor.

**Formal analysis:** Ucheoma Nwaozuru, Sarah Blackstone.

**Methodology:** Ucheoma Nwaozuru, Sarah Blackstone, Titi Gbajabiamila.

**Resources:** Patricia Iwelunmor.

**Supervision:** Titi Gbajabiamila, Oliver Ezechi, Patricia Iwelunmor, Juliet Iwelunmor.

**Validation:** Juliet Iwelunmor.

**Writing – original draft:** Ucheoma Nwaozuru, Sarah Blackstone, Chisom Obiezu-Umeh, Donaldson F. Conserve, Juliet Iwelunmor.

**Writing – review & editing:** Ucheoma Nwaozuru, Sarah Blackstone, Chisom Obiezu-Umeh, Donaldson F. Conserve, Stacey Mason, Florida Uzoaru, Titi Gbajabiamila, Oliver Ezechi, Patricia Iwelunmor, John E. Ehiri, Juliet Iwelunmor.

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
