## [Decision Letter · Decision Letter 0]

12 Dec 2019

PONE-D-19-29960

Psychosocial correlates of safe sex self-efficacy among in-school adolescent girls in Lagos, Nigeria

PLOS ONE

Dear Ms Nwaozuru,

Thank you for submitting your manuscript to PLOS ONE. After careful consideration, we feel that it has merit but does not fully meet PLOS ONE’s publication criteria as it currently stands. Therefore, we invite you to submit a revised version of the manuscript that addresses the points raised during the review process.

We would appreciate receiving your revised manuscript by Jan 26 2020 11:59PM. To enhance the reproducibility of your results, we recommend that if applicable you deposit your laboratory protocols in protocols.io, where a protocol can be assigned its own identifier (DOI) such that it can be cited independently in the future. For instructions see: http://journals.plos.org/plosone/s/submission-guidelines#loc-laboratory-protocols

We look forward to receiving your revised manuscript.

Kind regards,

Wendee Wechsberg

Academic Editor

PLOS ONE

Journal Requirements:

1. Please refer to any sample size calculations performed prior to your statistical analyses. If this was not performed please justify the reasons. Please refer to our statistical reporting guidelines for assistance (https://journals.plos.org/plosone/s/submission-guidelines.#loc-statistical-reporting). In addition, please include additional information regarding the survey or questionnaire used in the study and ensure that you have provided sufficient details that others could replicate the analyses. For instance, if you developed a questionnaire as part of this study and it is not under a copyright more restrictive than CC-BY, please include a copy, in both the original language and English, as Supporting Information.

3. Please include a copy of Table 4 which you refer to in your text on page 13.

<h3>**4.We note you have included a table to which you do not refer in the text of your manuscript. Please ensure that you refer to Table 5 in your text; if accepted, production will need this reference to link the reader to the Table.**</h3>

Reviewers' comments:

Reviewer's Responses to Questions

**Comments to the Author**

1. Is the manuscript technically sound, and do the data support the conclusions?

Reviewer #1: Yes

Reviewer #2: Partly

2. Has the statistical analysis been performed appropriately and rigorously? 

Reviewer #1: Yes

Reviewer #2: Yes

3. Have the authors made all data underlying the findings in their manuscript fully available?

Reviewer #1: Yes

Reviewer #2: Yes

4. Is the manuscript presented in an intelligible fashion and written in standard English?

Reviewer #1: Yes

Reviewer #2: Yes

5. Review Comments to the Author

Reviewer #1: Summary

This study examined the correlation between various psychosocial factors and safe sex self-efficacy among adolescent girls in Nigeria. The authors found that future orientation, participant age, and HIV knowledge were key correlates of safe sex self-efficacy.

Some of the strengths of the study is that it is among the first studies conducted on this topic with adolescent girls in Nigeria and it captured perspectives of younger adolescents (ages 13-15) which are less often included sexual health studies. Some of the weaknesses of this study include the low cronbach’s alphas on the measures used to assess future orientation and social support. Given that future orientation was one of the factors most strongly associated with safe sex self-efficacy, this is a notable weakness. Further, there was some redundancies in the text that need to be cut.

To improve the manuscript, I suggest the following changes:

Introduction

1) Some results are split by school type. Provide justification in the introduction for why school type is an important variable to stratify results by

Methods

2) Page 7, how was the survey administered? Paper and pencil? Interviewer administered?

3) Page 7, has the self-efficacy scale been used in Sub-Saharan Africa? What adaptations were made? Include the range of possible scores for the scale.

4) Page 8, the cronbachs alpha for the future orientation scale was quite low at .52. Was factor analysis conducted to examine if certain items should be removed from the scale. Additionally, include the range of possible scores for the scale.

5) MPSS cronbach’s alpha was .66 which is quite low. Again, was factor analysis conducted?

Results

6) Page 11, mentor definition should be moved to methods

7) Page 11, remove mean and standard deviations of age and self-efficacy in second paragraph

8) Page 12, the information presented in tables 2 and 3 are redundant given the nearly identical means and SDs for each variable. Cut table 2 and accompanying text.

9) Page 13, The differences in mean scores between public and private school are very small. Cut the sentences that include the means.

10) Page 14, table 4 mentioned in the first paragraph should be table 5

Discussion

11) Page 16, the sentence, “Therefore, increasing their confidence in safe sex self-efficacy, as with increased HIV knowledge in this study, was associated with increases in safe sex self-efficacy” was unclear.

12) Page 17, the paragraph that begins, “Taken together…” can be cut as it repeats content in the following paragraph.

Reviewer #2: This manuscript needed much improvement. If the issues are addressed it will be suitable for publication.

6. PLOS authors have the option to publish the peer review history of their article (what does this mean?). If published, this will include your full peer review and any attached files.

Reviewer #1: No

Reviewer #2: No

---

## [Author Response · Author response to Decision Letter 0]

26 Jan 2020

Editor 

PLOS ONE

January 24th, 2020 

Dear Editor,

We appreciate the time and effort that you and the reviewers have dedicated to providing your valuable feedback on the attached manuscript. We have been able to incorporate changes to reflect most of the suggestions provided by the reviewers. Here is a point-by-point response to the reviewers’ comments and concerns.

Journal Requirements:

1. Please refer to any sample size calculations performed prior to your statistical analyses. If this was not performed please justify the reasons. Please refer to our statistical reporting guidelines for assistance (https://journals.plos.org/plosone/s/submission-guidelines.#loc-statistical-reporting). In addition, please include additional information regarding the survey or questionnaire used in the study and ensure that you have provided sufficient details that others could replicate the analyses. For instance, if you developed a questionnaire as part of this study and it is not under a copyright more restrictive than CC-BY, please include a copy, in both the original language and English, as Supporting Information.

Response: We did not perform sample size calculations performed prior to the statistical analyses. However, we ascertained our sample size (N=426) adequate based on similar studies conducted among in-school adolescent girls in Nigeria that had sample size of 300 participants or less [1, 2]. We have also included a copy of the questionnaire as a supplementary file. 

Response: We will update your Data Availability statement on your behalf to reflect the information you provide. The URL to the data file is: https://www.openicpsr.org/openicpsr/workspace Accession number: openicpsr-117403

3. Please include a copy of Table 4 which you refer to in your text on page 13.

Response: Table 4 is now presented as Table 3 based on feedback from reviewer #1. Two tables were consolidated to one table. 

4.We note you have included a table to which you do not refer in the text of your manuscript. Please ensure that you refer to Table 5 in your text; if accepted, production will need this reference to link the reader to the Table.

Response: Thanks for noting this, we referred to Table 5 now Table 4 in the updated manuscript

“Psychosocial correlates of safe sex self-efficacy among in-school adolescent girls in Lagos, Nigeria" is a paper whose topic is likely to be of interest to the readership of PLoS One. The current study examined how social support, parental monitoring, and future orientation influence perceived safe sex self-efficacy among in-school adolescent girls in Nigeria. This study contributes to the literature by examining correlates of safe sex self-efficacy among a unique population. However, I have several concerns about the lack of information regarding the rationale for the study, the omission of sexual behavior data from the models, and the inferences made in the discussion. If these points of clarification can be addressed, I would recommend resubmitting to PLos One for publication.

Central Concerns:

1. My overarching concern for this paper involves its lack of rationale. While it is clear that adolescent girls and young women (AGYW) have high rates of HIV an STIs, it is not clear how this paper moves the research forward or what theoretical or conceptual base was used to develop the research question. 

 Response: Thanks for comment. We have edited the introduction section to 

 strengthen the rationale for the study and highlight the theoretical guide for the study 

 and variable selection. 

2. There has been much research done on correlates of sexual self-efficacy. How is the paper different? How are Nigerian AGYW different from other populations and why is this research needed?

Response: Thanks for this question. 

We included how cultural factors and economic disparities among adolescent girls and young women make them unique and increases their risk to engage in risky sexual behavior. 

3. What is the theoretical rationale for including all these measures in the models? It seems like the Theory of Planned Behavior or the Theory of Planned Behavior and Reasoned Action may be guiding this work, but the inclusion of social support and parental monitoring makes this unclear. Please clarify. 

Response: The theory guiding the study is reasoned action approach. This is introduced in paragraph 3 of the introduction

4. The fact that there is no data on whether or not the AGYW in the study were sexually active is a major limitation. Those who are sexually active are likely different than those who are not - especially in terms of safe sex self-efficacy. If this data was collected it should be entered as a covariate. If not, these findings are quite limited.

Response: Thank you for pointing this out. This has been highlighted in the limitation section:

“In addition, we did not ascertain which of the participants were sexually active”

Other points

5. In the introduction, the second paragraph is disjointed. The point of the paragraph is to discuss predictors of self-efficacy. Therefore, the section should first discuss why self-efficacy is important in predicting sexual behavior. Then discuss the associations between the other variables and self-efficacy. This section should end with a stronger rationale for examining these associations among AGYW in Nigeria.

Response: Thanks for pointing this out. We have reworked the introduction by discussing the importance of self-efficacy is important in predicting sexual behavior

6. Safe sex negotiation is different from self-efficacy. Therefore, the authors should either remove the reference to safe sex negotiation or integrate it into the conceptual framework and analysis. 

Response: Thanks for pointing this out. We deleted the reference to safe sex negotiation in the paper. 

7. The authors should elaborate on why each independent variable may be related to safe sex self-efficacy – placing emphasis on if these associations have been demonstrated among AGYW and/or AGYW in Nigeria. The introduction should also explain why it is important to look at these associations among AGYW in Nigeria.

Response: Thank you the suggestion. We have highlighted the importance of the assessing safe sex self-efficacy correlates among AGYW and AGYW in Nigeria. 

8. Please elaborate on why it is important for research to be conducted at schools (e.g., school-based interventions). 

Response: Thank you for the suggestion. We provided additional rationale for conducted research in Nigeria. 

“ Schools were selected because they provided an optimal opportunity to reach many adolescents. It is also the only primary institution with the most number of adolescent attendance, and a large number of adolescents attend schools before they are predisposed to sexual behaviors that may put them at risk of HIV infection [28].It is also an ideal platform to implement interventions that build confidence in health-seeking behaviors, thereby enhancing overall self-efficacy [29]. In Nigeria variations exist between the two main school systems (private and public schools). These schools differ by the availability of financial and human resources [30]. For instance, private schools are more likely to have access to funding because they are mostly for-profit. The present study assessed the psychosocial correlates of safer sex practices among in-school adolescent girls in Nigeria [31] . The findings from this study may inform the design of interventions that effectively consider the role psychosocial factors play in sexual decision-making and intentions and behaviors among adolescent girls in Nigeria.”

9. Please describe your randomization method.

Response: Thanks for the comment. This highlighted in the method section as: 

“The randomization was done at the district level. A list of schools in the two districts were obtained and stratified by type of school (i.e. public-owned and private-owned). We randomly selected 10 schools from each district using random numbers generated with MS Excel. We further reached out to the principals of the schools selected through the randomization process. Schools selected for the study”. 

10. Please describe how parental consent was obtained. 

Response: We included more information on how parental consent was obtained for the study in the methods section – Procedures. 

“Afterschool principals’ written approvals and review of study questionnaire, letters of intent, written in English, were distributed to adolescent girls in the twelve schools to notify their parents of the study. Parents and adolescents at the schools were informed that participation was voluntary, and that their responses to the study survey were confidential and anonymous. Students who were under 17 years and interested in the study were provided with a consent form to take home to their parents prior to the study date to obtain their parents’ consent”. 

11. Please describe how survey was administered (e.g., read aloud). 

Response: We have updated this in the methods section. 

“The questionnaires were self-administered and completed using pen or pencil.”

12. Please give the American, English, and other relevant equivalents to Senior Secondary School.

Response: Thanks for highlighting this. We included the America equivalent for the secondary school grades in the sentence below in the methods section. 

“Study participants were eligible to participate in the study if they were 13 years and above, attending secondary schools enrolled in JSS1, JSS2, JSS3, SS1, SS2, and SS3 (equivalent to Grades 6 through 12 in the United States)…”

13. Please include a meaningful cut-off or norm score for the measures in Table 2. For example, what score represents average knowledge of HIV?

Response: Thanks for the comment. We included have included the range scores for HIV knowledge and other scales in the study. Table 2 was deleted as per suggestion from Reviewer #1. 

14. Please clarify whether age was modeled as a continuous or categorical variable. It should be modeled as a continuous variable.

Response: Age was modeled as a continuous variable in the analysis. We specified this in the methods section. 

“Demographic Factors: We assessed participants’ age (continuous variable in the analyses), ethnicity (categorized as Igbo, Yoruba, Hausa and Others), educational level (categorized as JSS1, JSS2, JSS3, SS1, SS2, and SS3), religion (Categorized as Christianity or Islam), and whether they had a mentor (Yes or No response). This was defined as someone over 25 years that they could go to for support and guidance if they needed to make an important decision, or who inspired them to do their best. We also assessed school type (public or private schools).”

15. Typically, variables that are not related to the outcome in the bivariate analysis are not included in the multivariate model. Therefore, please provide a rationale for including all the demographic and psychosocial factors in the models.

Response: The rationale for including the demographic and psychosocial factors in the model is based on previous studies that suggest associations between these factors and safe sex self-efficacy. We wanted to assess the extent to which this would hold true in our study sample. 

16. The association between religion and self-efficacy is an interesting finding. Perhaps you should conduct the bivariate analysis separated by type of school? No table would be necessary. It would just be to identify important correlates of each model.

Response: Thank you for the suggestion. A bivariate analysis between religion and safe sex self-efficacy was not statistically significant. However, when the bivariate analysis between religion and safe sex self-efficacy was stratified by school type. The association between religion and safe sex self-efficacy was statistically significant for participants in the public school (t=20.70 (df=60), p=0.03) but not statistically significant for participants in private schools (t=38.40 (df=141), p=0.10). 

17. In the discussion the authors state:

“Thus, while HIV knowledge may not be associated with risky sexual behaviors, it could potentially be indirectly associated with risky sexual behaviors and mediated by safe sex self-efficacy. While this study did not examine actual sexual behavior, this potential mediating relationship should be considered in future studies. 

This mediating relationship is the impetus for several theories. Perhaps the authors should describe how this relationship may be different for this population.

Response: This relationship may be different for adolescent girls and young women given the complex interplay of cultural and social factors that increase their risk of sexual risky behaviors. 

18. The findings related to future orientation are quite important! It is also important for interventions to understand future orientation for HIV prevention interventions.

Response: Thank you for the comment. 

19. “Protective” language is too strong without having conducted a moderation analyses - see Rutter's work on resilience

Response: Thank you for this comment. We have deleted “protective” in the statement. 

Reviewer #1: Summary

This study examined the correlation between various psychosocial factors and safe sex self-efficacy among adolescent girls in Nigeria. The authors found that future orientation, participant age, and HIV knowledge were key correlates of safe sex self-efficacy.

Some of the strengths of the study is that it is among the first studies conducted on this topic with adolescent girls in Nigeria and it captured perspectives of younger adolescents (ages 13-15) which are less often included sexual health studies. Some of the weaknesses of this study include the low Cronbach’s alphas on the measures used to assess future orientation and social support. Given that future orientation was one of the factors most strongly associated with safe sex self-efficacy, this is a notable weakness. Further, there was some redundancies in the text that need to be cut.

To improve the manuscript, I suggest the following changes:

Introduction

1) Some results are split by school type. Provide justification in the introduction for why school type is an important variable to stratify results by Response: Thank you for point this out. We updated the introduction section to highlight the justification for stratifying the results by school type. 

“ Schools were selected because they provided an optimal opportunity to reach many adolescents. It is also the only primary institution with the most number of adolescent attendance, and a large number of adolescents attend schools before they are predisposed to sexual behaviors that may put them at risk of HIV infection [28].It is also an ideal platform to implement interventions that build confidence in health-seeking behaviors, thereby enhancing overall self-efficacy [29]. In Nigeria variations exist between the two main school systems (private and public schools). These schools differ by the availability of financial and human resources [30]. For instance, private schools are more likely to have access to funding because they are mostly for-profit. The present study assessed the psychosocial correlates of safer sex practices among in-school adolescent girls in Nigeria [31] . The findings from this study may inform the design of interventions that effectively consider the role psychosocial factors play in sexual decision-making and intentions and behaviors among adolescent girls in Nigeria.”

Methods 2) Page 7, how was the survey administered? Paper and pencil? Interviewer administered? Response: We have updated this. The questionnaires were self-administered and completed using pen or pencil.

3) Page 7, has the self-efficacy scale been used in Sub-Saharan Africa? What adaptations were made? Include the range of possible scores for the scale. Response: Some statements from the scale have been used among adolescents in sub-Saharan Africa. The adaptation we made by rewording some of the sentences to enhance clarity based on feedback from some young people. We included the possible range of scores for the scales in the study. 

4) Page 8, the cronbachs alpha for the future orientation scale was quite low at .52. Was factor analysis conducted to examine if certain items should be removed from the scale. Additionally, include the range of possible scores for the scale.

Response: Thanks for the comment. We conducted an exploratory factor analysis (principal components) and removing of items from the scale did not improve the reliability of the scale for our sample. Therefore, we decided to leave the items to avoid interfering with validity and credibility of the scale. Since it was already previously validated. We also included this statement for scales with Cronbach alpha’s lower than 0.7 “In no case did the deletion of an item increase the α by 0.01”. 

5) MPSS cronbach’s alpha was .66 which is quite low. Again, was factor analysis conducted? Response: Similar to the future orientation, we conducted an exploratory factor analysis (principal components) and removing of items from the scale did not improve the reliability of the scale for our sample. Therefore, we decided to leave the items to avoid interfering with validity and credibility of the scale. Since it was already previously validated. 

Results

6) Page 11, mentor definition should be moved to methods Response: Thanks. We deleted the definition of mentor in the results section. 

7) Page 11, remove mean and standard deviations of age and self-efficacy in second paragraph Response: We removed mean and standard deviations of age and self-efficacy in second paragraph of page 11 

8) Page 12, the information presented in tables 2 and 3 are redundant given the nearly identical means and SDs for each variable. Cut table 2 and accompanying text. Response: Thanks for the feedback. Table 2 provides on the independent variables for the entire participants. While Table 3 presents data on the independent variables by school type. We consolidated the table to create one table, which is now Table 2. Table 2 present the total sample mean for the variables and means by schools types. 

9) Page 13, The differences in mean scores between public and private school are very small. Cut the sentences that include the means. Response: We have deleted the mean scores as suggested since it is presented in Table 3. 

10) Page 14, table 4 mentioned in the first paragraph should be table 5 Response: Thanks, we have edited the paragraph to refer to table 5. Since we consolidated two tables, it now table 4 

Discussion

11) Page 16, the sentence, “Therefore, increasing their confidence in safe sex self-efficacy, as with increased HIV knowledge in this study, was associated with increases in safe sex self-efficacy” was unclear. Response: We reworded this sentence to say “This may have increased their confidence in practicing safe sex, as increased HIV knowledge in this study was associated with increases in safe sex self-efficacy”.

12) Page 17, the paragraph that begins, “Taken together…” can be cut as it repeats content in the following paragraph. Response: Thank for point this out. We have deleted this paragraph.

We hope the revised manuscript will better suit the journal, but we are happy to make further revisions if needed.

Sincerely, 

Ucheoma Nwaozuru, MS

1. Nwokocha AR, Nwakoby BA. Knowledge, attitude, and behavior of secondary (high) school students concerning HIV/AIDS in Enugu, Nigeria, in the year 2000. Journal of Pediatric and Adolescent Gynecology. 2002;15(2):93-6.

2. Ajide KB, Balogun FM. Knowledge of HIV and intention to engage in risky sexual behaviour and practices among senior school adolescents in Ibadan, Nigeria. Archives of basic and applied medicine. 2018;6(1):3.

---

## [Decision Letter · Decision Letter 1]

27 Feb 2020

PONE-D-19-29960R1

Psychosocial correlates of safe sex self-efficacy among in-school adolescent girls in Lagos, Nigeria

PLOS ONE

Dear Ms Nwaozuru,

Thank you for submitting your manuscript to PLOS ONE. After careful consideration, we feel that it has merit but does not fully meet PLOS ONE’s publication criteria as it currently stands. Therefore, we invite you to submit a revised version of the manuscript that addresses the points raised during the review process.

We would appreciate receiving your revised manuscript by Apr 12 2020 11:59PM. To enhance the reproducibility of your results, we recommend that if applicable you deposit your laboratory protocols in protocols.io, where a protocol can be assigned its own identifier (DOI) such that it can be cited independently in the future. For instructions see: http://journals.plos.org/plosone/s/submission-guidelines#loc-laboratory-protocols

We look forward to receiving your revised manuscript.

Kind regards,

Wendee Wechsberg

Academic Editor

PLOS ONE

Reviewers' comments:

Reviewer's Responses to Questions

**Comments to the Author**

1. If the authors have adequately addressed your comments raised in a previous round of review and you feel that this manuscript is now acceptable for publication, you may indicate that here to bypass the “Comments to the Author” section, enter your conflict of interest statement in the “Confidential to Editor” section, and submit your "Accept" recommendation.

Reviewer #1: (No Response)

Reviewer #2: All comments have been addressed

2. Is the manuscript technically sound, and do the data support the conclusions?

Reviewer #1: Yes

Reviewer #2: Yes

3. Has the statistical analysis been performed appropriately and rigorously? 

Reviewer #1: Yes

Reviewer #2: Yes

4. Have the authors made all data underlying the findings in their manuscript fully available?

Reviewer #1: Yes

Reviewer #2: Yes

5. Is the manuscript presented in an intelligible fashion and written in standard English?

Reviewer #1: Yes

Reviewer #2: Yes

6. Review Comments to the Author

Reviewer #1: (No Response)

Reviewer #2: While, the authors have addressed the comments of the reviewers. The manuscript needs to be thoroughly edited. Furthermore, the introduction of the manuscript still needs substantial improvement.

The first paragraph is entirely too long and disjointed. The authors should redevelop the introduction so that it comprises separate and coherent paragraphs with the following:

Paragraph 1 -introduction to the problem

Paragraph 2 - Previous research examining the problem

Paragraph 3 - Theoretical Framework for understanding the problem

Paragraph 4 - Current study

This explanation of the RAA is a great start. However, the authors should elaborate on the major points of the RAA and then situate the constructs of the current study in this framework.

7. PLOS authors have the option to publish the peer review history of their article (what does this mean?). If published, this will include your full peer review and any attached files.

Reviewer #1: No

Reviewer #2: No

---

## [Author Response · Author response to Decision Letter 1]

12 Apr 2020

School for Public Health and Social Justice 

Salus Center 

3545 Lafayette Avenue 

Saint Louis, MO 63103

Email: nwaozur2@gmail.com

Dear Dr. Wendee Wechsberg, 

We appreciate the time and effort that you and the reviewers have dedicated to providing additional feedback to the manuscript: Psychosocial correlates of safe sex self-efficacy among in-school adolescent girls in Lagos, Nigeria (PONE-D-19-29960R1). We have been able to incorporate changes to reflect most of the suggestions provided by the reviewers. Here is a point-by-point response to the reviewers’ comments and concerns (please note that responses to the comments are in BOLD). 

Comments to the Author

1. If the authors have adequately addressed your comments raised in a previous round of review and you feel that this manuscript is now acceptable for publication, you may indicate that here to bypass the “Comments to the Author” section, enter your conflict of interest statement in the “Confidential to Editor” section, and submit your "Accept" recommendation.

Reviewer #1: (No Response)

Reviewer #2: All comments have been addressed

Response: Thank you for the kind response. 

2. Is the manuscript technically sound, and do the data support the conclusions?

Reviewer #1: Yes

Reviewer #2: Yes

Response: Thank you for the kind response. 

3. Has the statistical analysis been performed appropriately and rigorously?

Reviewer #1: Yes

Reviewer #2: Yes

Response: Thank you for the kind response. 

4. Have the authors made all data underlying the findings in their manuscript fully available?

Reviewer #1: Yes

Reviewer #2: Yes

Response: Thank you for the kind response. 

 5. Is the manuscript presented in an intelligible fashion and written in standard English?

Reviewer #1: Yes

Reviewer #2: Yes

Response: Thank you for the kind response. 

6. Review Comments to the Author

Reviewer #1: (No Response)

Reviewer #2: While, the authors have addressed the comments of the reviewers. The manuscript needs to be thoroughly edited. Furthermore, the introduction of the manuscript still needs substantial improvement.

The first paragraph is entirely too long and disjointed. The authors should redevelop the introduction so that it comprises separate and coherent paragraphs with the following:

Paragraph 1 -introduction to the problem

Paragraph 2 - Previous research examining the problem

Paragraph 3 - Theoretical Framework for understanding the problem

Paragraph 4 - Current study

This explanation of the RAA is a great start. However, the authors should elaborate on the major points of the RAA and then situate the constructs of the current study in this framework.

We appreciate the helpful feedback. We have revised the manuscript introduction to reflect the reviewer’s feedback. We restructured the introduction to be concise and include the following information: introduction to the problem, previous research examining the problem, theoretical framework for understanding the problem and the current study. 

7. PLOS authors have the option to publish the peer review history of their article (what does this mean?). If published, this will include your full peer review and any attached files.

Do you want your identity to be public for this peer review? For information about this choice, including consent withdrawal, please see our Privacy Policy.

Reviewer #1: No

Reviewer #2: No

Sincerely, 

Ucheoma Nwaozuru, MSc

---

## [Decision Letter · Decision Letter 2]

3 Jun 2020

Psychosocial correlates of safe sex self-efficacy among in-school adolescent girls in Lagos, Nigeria

PONE-D-19-29960R2

Dear Dr. Nwaozuru,

We’re pleased to inform you that your manuscript has been judged scientifically suitable for publication and will be formally accepted for publication once it meets all outstanding technical requirements.

Kind regards,

Wendee Wechsberg

Academic Editor

PLOS ONE

Additional Editor Comments (optional):

Reviewers' comments:

Reviewer's Responses to Questions

**Comments to the Author**

1. If the authors have adequately addressed your comments raised in a previous round of review and you feel that this manuscript is now acceptable for publication, you may indicate that here to bypass the “Comments to the Author” section, enter your conflict of interest statement in the “Confidential to Editor” section, and submit your "Accept" recommendation.

Reviewer #1: All comments have been addressed

Reviewer #2: All comments have been addressed

2. Is the manuscript technically sound, and do the data support the conclusions?

Reviewer #1: Yes

Reviewer #2: Yes

3. Has the statistical analysis been performed appropriately and rigorously? 

Reviewer #1: Yes

Reviewer #2: Yes

4. Have the authors made all data underlying the findings in their manuscript fully available?

Reviewer #1: Yes

Reviewer #2: Yes

5. Is the manuscript presented in an intelligible fashion and written in standard English?

Reviewer #1: Yes

Reviewer #2: Yes

6. Review Comments to the Author

Reviewer #1: (No Response)

Reviewer #2: The authors have adequately addressed the critiques of the reviewers. The authors have done a good job of restructuring the introduction to be concise and including the introduction to the problem, previous research examining the problem, theoretical framework for understanding the problem and the current study.

7. PLOS authors have the option to publish the peer review history of their article (what does this mean?). If published, this will include your full peer review and any attached files.

Reviewer #1: No

Reviewer #2: No

---

## [Editor Report · Acceptance letter]

9 Jun 2020

PONE-D-19-29960R2 

Psychosocial correlates of safe sex self-efficacy among in-school adolescent girls in Lagos, Nigeria 

Dear Dr. Nwaozuru:

I'm pleased to inform you that your manuscript has been deemed suitable for publication in PLOS ONE. Congratulations! Your manuscript is now with our production department. 

Kind regards, 

on behalf of

Dr. Wendee Wechsberg 

Academic Editor

PLOS ONE